# Recombinant Anti-PF4 Antibodies Derived from Patients with Vaccine-Induced Immune Thrombocytopenia and Thrombosis (VITT) Facilitate Research and Laboratory Diagnosis of VITT

**DOI:** 10.3390/vaccines13010003

**Published:** 2024-12-24

**Authors:** Luisa Müller, Venkata A. S. Dabbiru, Lucy Rutten, Rinke Bos, Roland Zahn, Stefan Handtke, Thomas Thiele, Marta Palicio, Olga Esteban, Marta Broto, Tom Paul Gordon, Andreas Greinacher, Jing Jing Wang, Linda Schönborn

**Affiliations:** 1Institut für Transfusionsmedizin, Universitätsmedizin Greifswald, 17489 Greifswald, Germany; luisa.mueller@med.uni-greifswald.de (L.M.); venkata.dabbiru@med.uni-greifswald.de (V.A.S.D.); stefan.handtke@med.uni-greifswald.de (S.H.); thomas.thiele@med.uni-greifswald.de (T.T.); andreas.greinacher@med.uni-greifswald.de (A.G.); 2Janssen Vaccines & Prevention BV, 2333 CN Leiden, The Netherlands; lrutten@its.jnj.com (L.R.); rbos6@its.jnj.com (R.B.); rzahn@its.jnj.com (R.Z.); 3Werfen, Lliçà d’Amunt, 08186 Barcelona, Spain; mpalicio@werfen.com (M.P.); mbroto@werfen.com (M.B.); 4Department of Immunology, College of Medicine and Public Health, Flinders University and SA Pathology, Bedford Park, Adelaide, SA 5042, Australia; t.gordon@flinders.edu.au (T.P.G.); jingjing.wang@flinders.edu.au (J.J.W.)

**Keywords:** vaccine-induced immune thrombocytopenia and thrombosis (VITT), platelet-factor 4, recombinant antibodies, platelet activation assay

## Abstract

Background/Objectives: Adenoviral vector-based vaccines against COVID-19 rarely cause vaccine-induced immune thrombocytopenia and thrombosis (VITT), a severe adverse reaction caused by IgG antibodies against platelet factor 4 (PF4). To study VITT, patient samples are crucial but have become a scarce resource. Recombinant antibodies (rAbs) derived from VITT patient characteristic amino acid sequences of anti-PF4 IgG are an alternative to study VITT pathophysiology. Methods: Amino acid sequences of the variable region of immunoglobulin light and heavy chain of anti-PF4 IgG derived from VITT patients were obtained by mass spectrometry sequencing and rAbs were synthetized by reverse-engineering. Six different rAbs were produced: CR23003, CR23004, and CR23005 (from a patient vaccinated with Jcovden, Johnson & Johnson-Janssen (Beerse, Belgium)), CR22046, and CR22050 and CR22066 (from two different patients vaccinated with Vaxzevria, AstraZeneca (Cambridge, UK)). These rAbs were further characterized using anti-PF4 and anti-PF4/heparin IgG ELISAs, rapid anti-PF4 and anti-PF4/polyanion chemiluminescence assays, and PF4-induced platelet activation assay (PIPA) and their capacity to induce procoagulant platelets. Results: rAbs bound to PF4 alone, but not to PF4/polyanion complexes in rapid chemiluminescence assays. Chemiluminescence assays and both anti-PF4 IgG and anti-PF4 IgG/heparin ELISA showed concentration-dependent PF4 binding of all six rAbs, however, with different reactivities among them. PIPA showed a similar, concentration-dependent platelet activation pattern. rAbs varied in their reactivity and the majority of the tested rAbs were able to induce procoagulant platelets. Conclusions: The six rAbs derived from VITT patients reflect VITT-typical binding capacities and the ability to activate platelets. Therefore, these rAbs offer an attractive new option to study VITT pathophysiology.

## 1. Introduction

In response to the coronavirus disease 2019 (COVID-19) pandemic, a massive global vaccination campaign was carried out. Different types of vaccines were developed rapidly, including adenoviral vector-based vaccines like ChAdOx1 nCoV-19 (Vaxzevria, AstraZeneca) and Ad26.COV2.S (Jcovden, Johnson & Johnson-Janssen) [1]. Adenoviral vector-based vaccines cause rare adverse effects in a small fraction of individuals that are mainly characterized by thrombosis and thrombocytopenia [2,3,4,5], termed vaccine-induced immune thrombocytopenia and thrombosis (VITT). The incidence differs depending on the reporting country and the vaccine used. The incidence ranges from 1 case per 26,500 to 127,300 for the first doses of ChAdOx1 nCoV-19 and has been estimated as 1 case per 263,000 Ad26.COV2.S doses administered in the US [3]. VITT occurs as a secondary immune reaction typically 5–30 days after vaccination and is clinically represented by reduced platelet counts, strongly elevated D-dimer levels, platelet-activating anti-platelet factor 4 (PF4) immunoglobulin G (IgG) antibodies, and thrombosis, especially at atypical sites such as the cerebral venous sinus and/or splanchnic veins [6].

VITT resembles another antibody-mediated prothrombotic disorder called heparin-induced thrombocytopenia (HIT), but in contrast to HIT antibodies, VITT antibodies are slightly different and are not induced by heparin. Both VITT and HIT antibodies bind to PF4 but recognize different epitopes [7]. VITT antibodies strongly bind to PF4 alone, while most HIT antibodies only bind to PF4/heparin (or other polyanion) complexes [8]. This has important implications for the laboratory testing of HIT and VITT antibodies [2]. In microtiter plate-based anti-PF4/heparin assays, both types of antibodies bind [9,10]. Therefore, these assays do not allow differentiation between anti-PF4/heparin (typical for HIT) and anti-PF4 antibodies (typical for VITT). The combination of two anti-PF4 antibody chemiluminescence assays now allows the differentiation of these different binding characteristics [11].

Shortly after the characterization of VITT, vaccination with adenoviral vector-based vaccines was stopped in many countries. This has effectively reduced the number of VITT cases; however, this also severely limits the availability of patient-derived material (i.e., sera) for the research of the underlying pathomechanisms of VITT [12]. Clarifying the underlying mechanism of VITT is not only desirable for the clinical management of patients with anti-PF4 disorders, but also holds crucial implications for future vaccine development. Adenoviral vector platforms are very promising, and at the same time, production costs are considerably lower and required storage conditions more favorable than, e.g., for mRNA-based vaccines. This is very important for the development of affordable vaccines for almost neglected diseases, especially in low- and middle-income countries [13]. Reducing or even abandoning the risk for VITT would substantially increase adenoviral-based vaccine safety.

A possible solution to overcome the limitation of the scarcity of patient-derived antibodies is the production of recombinant antibodies (rAbs), which mimic VITT antibodies in patient sera [12]. The production of human-derived antibodies usually depends on the availability of specific B-cells in patients [14,15] to isolate the gene sequence of the hypervariable region. However, barely any B-cells of VITT patients are available from the acute phase. A very promising new approach for obtaining such rAbs in the absence of available, patient-derived B-cells is based on proteomic analysis. For this approach, anti-PF4 antibodies from VITT patients’ sera were affinity purified using PF4 bound to a solid phase. Then, the amino acid sequences of the immunoglobulin variable region of the light and heavy chains were sequenced and used to generate DNA to express full-length IgGs with the same antigen recognition sites as the patients’ antibodies [16].

In this article, we characterize six of these rAbs derived from three different patients with classic VITT after vaccination with ChAdOx1 nCoV-19 (Vaxzevria, AstraZeneca) or Ad26.COV2.S (Jcovden, Johnson & Johnson-Janssen) vaccine. The rAbs were characterized using anti-PF4/heparin and anti-PF4 enzyme-linked immunosorbent assay (ELISA), a rapid chemiluminescence assay for anti-PF4/polyanion and anti-PF4 antibodies, platelet activation assays, and the ability of rAbs to induce procoagulant platelets, using current standard approaches for VITT diagnosis in specialized laboratories [17,18].

## 2. Materials and Methods

### 2.1. Patient Samples and Antibody Generation

Serum samples with suitable volumes of three VITT patients from the acute phase (obtained 7–15 days after vaccination) were used to affinity-purify serum anti-PF4 antibodies as previously described [19]. The serum samples of patient P1 (VITT after vaccination with Ad26.COV2.S, Jcovden, Johnson & Johnson-Janssen) were collected at Universitätsmedizin Greifswald, Germany, and those of patient P2 and patient P3 (VITT after vaccination with ChAdOx1 nCoV-19, Vaxzevria, AstraZeneca) at Flinders Medical Centre, Australia (approved by the Clinical Ethics Committees the Universitätsmedizin Greifswald [approval number BB052/21a] and of the Flinders Medical Centre [approval number 39.034]). Data of P1 were previously published as a study subject of the German VITT cohort [20], data of P2 were previously published as VITT2, and of P3 as VITT5 [19].

Originating from affinity-purified anti-PF4 antibodies, rAbs were produced, as described [16]. After the amino acid sequences of the hypervariable regions of the antibodies had been obtained by mass spectrometric sequencing, the sequence was transcribed into DNA. DNA fragments encoding full-length IgG antibody heavy chain (HC) and light chain (LC) were synthesized and cloned into pcDNA3.4 vector separately, in a frame with an artificial signal peptide in front. The resulted constructs were co-transfected into CHO cells at an HC:LC ratio of 1:1. The culture medium supernatant was harvested by ultrafiltration, and then subjected to Protein A affinity purification and SEC-HPLC chromatography polishing. Antibody purity was characterized by SDS-PAGE. The purified antibodies were dialyzed into 20 mM NaAC, 75 mM NaCl, 5% sucrose, and pH 5.5, and stored at −80 °C until required. Production was carried out at a large scale by a subcontractor. From the sequences obtained from the three VITT-patients, in total six different PF4-binding antibodies were obtained. Based on these slightly different sequences, six rAbs were produced (three different rAbs derived from patient P1, one rAb derived from patient P2, and two different rAbs derived from patient P3).

### 2.2. Enzyme-Linked Immunosorbent Assay (ELISA)

CovaLink microtiter plate (Cat. No.: 10499762, ThermoFisher Scientific, Rosklide, Denmark) was coated with 100 µL of either 20 µg/mL PF4 (PF4-h, Chromatec, Greifswald, Germany) or 20 µg/mL PF4 (PF4-h, Chromatec, Greifswald, Germany) complexed with 0.5 IU/mL heparin (Heparin-Natrium, Ratiopharm, Ulm, Germany) in 50 mM NaH_2_PO_4_xH_2_O + 15 mM NaN_3_ at pH 7.5 at 4 °C for two weeks for anti-PF4 IgG ELISA and anti-PF4/heparin ELISA, respectively. For both assays, the plate was washed five times (0.1% Tween 20 in 0.15 M NaCl). Then, the rAbs were diluted (30 µg/mL, 15 µg/mL, 7.5 µg/mL, 3.75 µg/mL, 1.875 µg/mL, 0.937 µg/mL, 0.468 µg/mL, 0.234 µg/mL, and 0.177 µg/mL) in a dilution buffer (50 mM NaH_2_PO_4_xH_2_O + 0.15 M NaCl + 7.5% goat serum) and added to the plate for 1 h at room temperature. After washing five times with washing buffer, peroxidase-AffiniPure goat anti-human IgG, Fc-gamma fragment specific (109-036-098, Dianova, Hamburg, Germany; 1:15,000) was added. After 1 h of incubation, the plates were washed with washing buffer and 100 µL TMB-One liquid substrate (Cat. No. 4380, Kementec, Taastrup, Denmark) was applied for 10 min at room temperature and stopped with 100 µL of 0.5 M H_2_SO_4_. Absorbance at 450 nm was quantified on a BioChrom EZRead400 microplate reader (Tecan, Männedorf, Switzerland). The optical density (OD) of the negative control (well with dilution buffer but without any rAb) was subtracted from the raw OD values.

### 2.3. Chemiluminescence Assay for Anti-PF4 Antibodies

Two rapid assays were used to detect anti-PF4/polyanion- and anti-PF4-specific binding of the rAbs. We used the HemosIL AcuStar HIT-IgG (AcuStar HIT-IgG_(PF4-H)_ against PF4/polyvinyl sulfonate (PVS) complexes), hereafter called rapid anti-PF4/polyanion assay and an anti-PF4 antibody assay prototype for the ACL AcuStar, hereafter called rapid anti-PF4 assay (all from Werfen, Barcelona, Spain) [11]. Both assays are two-step chemiluminescence immunoassays consisting of magnetic particles coated either with PF4 complexed to PVS or with PF4 alone. rAbs bind to the PF4 or PF4/PVS coated magnetic particles and are detected using an anti-human IgG labeled with isoluminol. Both assays are used in the ACL AcuStar chemiluminescence analyzer (from Werfen, Barcelona, Spain). The rapid anti-PF4 assay was designed to detect antibodies that only recognize PF4 (as seen in VITT) with no or minimal cross-reactivity with anti-PF4/polyanions (typically seen in HIT). In the commercially available rapid anti-PF4/polyanion assay, the results are usually shown as U/mL. Because no units have yet been established for the new rapid anti-PF4 assay, the results for both assays are provided as generic Relative Light Units (RLUs) for better comparability. The cut-off for the rapid anti-PF4/heparin assay was calculated using calibrators, as per the manufacturer’s instructions. The cut-off of the new rapid anti-PF4 assay was determined by a receiver operating characteristic curve analysis between VITT and control samples elsewhere [10]. In the chemiluminescence assays, the antibodies were tested at different dilutions in a buffer supplied by the manufacturer starting at 30 µg/mL. Lower concentrations (15 µg/mL, 7.5 µg/mL, 3.75 µg/mL, 1.875 µg/mL, 0.937 µg/mL, 0.468 µg/mL, and 0.234 µg/mL) were only measured if the result of the previous dilution was still above the cut-off. To assess for low avidity cross-reactivity, the antibodies were also tested at higher concentrations of 60 µg/mL, 120 µg/mL, and 240 µg/mL in the rapid anti-PF4/polyanion chemiluminescence assay.

### 2.4. PF4-Induced Platelet Activation Assay with Washed Platelets (PIPA Test)

Platelet activation and aggregation by VITT patient-derived rAbs were tested as previously described [2], however in slightly lowered volumes. In brief, the washed platelets of healthy donors were incubated in the absence or presence of PF4 (PF4-h, Chromatec, Greifswald, Germany) with the patient-derived rAbs. A total of 35 µL platelet suspension was mixed with 10 µL rAbs or human IgG isotype control (Invitrogen, Rockford, IL, USA) in different concentrations (240 µg/mL, 120 µg/mL, 60 µg/mL, 30 µg/mL, 15 µg/mL, 7.5 µg/mL, 3.75 µg/mL, 1.875 µg/mL, 0.937 µg/mL, 0.468 µg/mL, 0.234 µg/mL, and 0.117 µg/mL). Additionally, either 6 µL PF4 (100 µg/mL) or 6 µL PBS (buffer control) was added. In addition, FcγRII-specific reaction was tested by the preincubation of the platelets with monoclonal antibody IV.3, which blocks FcγRIIa (cell supernatant of cell line ATCCHB-217, Biometec GmbH, Greifswald, Germany) 30 min before the PIPA. Incubation was performed under stirring conditions. Time to aggregation was measured up to 45 min. The test was determined to be positive if platelets aggregated within 30 min.

As human serum contains proteins that interfere with PF4 and PF4-platelet interaction, e.g., fibronectin [21], rAb solutions were diluted with normal human serum to compare them within the same matrix as VITT patients’ antibodies.

### 2.5. Flow Cytometry

To study the ability of the rAbs to cause a procoagulant state in platelets on the addition of PF4, flow cytometry was performed using the washed platelets of six different healthy donors. The washed platelets were preincubated with ReoPro (1:500, Eli Lilly, Giessen, Germany, blocking the platelet fibrinogen receptor) for 5 min to avoid platelet aggregation that would compromise the subsequent measurements. To 35 µL pretreated platelets, 10 µL rAbs (30 µg/mL) or human IgG isotype control (30µg/mL, Cat. No.: 12000C, Invitrogen, Rockford, IL, USA) and 6µL PF4 (100 µg/mL, PF4-h, Chromatec, Greifswald, Germany) were added and incubated under shear stress. Shear stress was applied for 5 min using two magnetic stirring beads added to each well of a 96-well plate placed on a magnetic laboratory stirrer (IKA C-MAG MS 7 at speed 2 = ~500 rpm). Then, an antibody master mix was added to the platelets, consisting of 68.4 µL Annexin V binding buffer (Cat. No.: 422201, BioLegend, Amsterdam, The Netherlands), 1.6 µL Hirudin 5400 U, 5 µL Annexin V (Cat. No.: A35110, APC-labelled, BioLegend, Amsterdam, The Netherlands), 5 µL CD62P-FITC (Cat. No: A07790, Beckmann Coulter, Villepinte, France), and 10 µL CD42b-PE (Cat. No: IM1417U, Beckmann Coulter, Villepinte, France). Platelets were incubated for 20 min at room temperature in the dark and measured using CytoFlexS (Beckmann Coulter, Villepinte, France). Events were pre-gated for the platelet marker CD42b, and platelets positive for both CD62P and for the phosphatidylserine marker Annexin V were quantified.

### 2.6. Statistical Analysis

Statistical analysis and data visualization were performed using GraphPad Prism (V 8.0.1, for Windows, GraphPad Software, Boston, MA, USA). Test for normal Gaussian distribution was performed using the Shapiro–Wilk test and subsequent suitable statistical testing. Outliers identified with the built-in ROUT method (Robust regression and Outlier removal) of GraphPad Prism with a maximum desired false discovery rate of 1% (Q = 1%) were removed from the data set. For detailed information, please see the Results Section.

## 3. Results

### 3.1. Patient Characteristics

The rAbs were obtained from the sequences of anti-PF4 antibodies found in the serum samples of patients with VITT. Their demographic and clinical findings are summarized in Table 1. Different rAbs derived from the same patient showed different variable region amino acid replacement mutations.

### 3.2. rAb Characterization

#### 3.2.1. PF4-Binding Capability of Patient-Derived rAbs

All antibodies bound in the anti-PF4 IgG ELISA (Figure 1a) and in the anti-PF4/heparin IgG ELISA (Figure 1b). In the chemiluminescence assays, antibodies bound to PF4 (Figure 1c) but rAbs reacted below the cut-off in the rapid anti-PF4/polyanion assay, however, with slight reactivity of CR23004, CR23005, and CR22050 at high concentrations (Figure 1d).

Overall, rAbs show concentration-dependent binding in the anti-PF4 IgG ELISA (Figure 1a, ordinary 2-way ANOVA, F_(7, 35)_ = 93.40, *p* < 0.0001) and in the anti-PF4/heparin IgG ELISA (Figure 1b, ordinary 2-way ANOVA, F_(7, 35)_ = 159.3, *p* < 0.0001). Similarly, the rAbs also behave in a concentration-dependent way in binding to PF4 in the rapid anti-PF4 assay (Figure 1c, 2-way ANOVA, rows with missing data were excluded, F_(4, 20)_ = 14.55, *p* < 0.0001). A major difference among these antibodies is seen in the rapid anti-PF4/polyanion ELISA. The rAbs CR23004, CR23005, and CR22050 show binding at high concentrations (Figure 1d, 2-way ANOVA, F_(10, 20)_ = 29.51, *p* < 0.0001). The rAb CR23003, CR22046, and CR22066 did not show binding in the rapid PF4/polyanion assay.

#### 3.2.2. Platelet Activation Assays with Patient-Derived rAbs

All the rAbs activated platelets in a concentration-dependent manner (Figure 2a, ordinary 2-way ANOVA, F_(11, 516)_ = 79.35, *p* < 0.0001) in the PIPA test with added PF4. Activation capability differed among the antibodies, with activation thresholds between 15 µg/mL and 60 µg/mL (Figure 2b). However, the differences in activation capability were not statistically significant. Using human IgG isotype as a negative control, no reactivity in the test was observed.

In the absence of exogenously added PF4, very limited activation of platelets was observed for the rAbs CR23004, CR23005, and CR22066 that activated platelets, but only at the highest tested concentration of 240 µg/mL (Appendix A). All the antibodies activated platelets via FcγRIIa as the monoclonal antibody IV.3, which blocks FcγRIIa, and inhibited the reaction (Appendix A).

#### 3.2.3. Induction of Procoagulant Platelets by the rAbs

Procoagulant platelets expressing phosphatidylserine (Annexin V) and P-selectin (CD62P) on their surface are a subpopulation of activated platelets, relevant for catalyzing thrombin generation. All the rAbs induced procoagulant platelets when compared to the signal in platelets treated with human IgG as a negative control (Figure 3, F_(6, 24.29)_ = 5.162, *p* = 0.0015). Except for rAb CR23003, the potential to induce procoagulant platelets was statistically significant for all the rAbs.

## 4. Discussion

In this study, we characterized six recombinant anti-PF4 rAbs generated from VITT patients’ sera. The rAbs react comparably to the patient sera in antigen assays and functional assays for anti-PF4 antibodies, as previously described [18,22,23]. However, there are differences among these antibodies. At the same concentration, they show different binding to PF4 and/or platelet activation. In addition, three of these antibodies also react with PF4/polyanion complexes, albeit at high concentrations only. In antigen assays (ELISA and chemiluminescence), differential binding depends on the antigen recognition site, while in functional assays, the reactivity of rAbs depends on both the antigen recognition site and the effector part of the antibody, the Fc part. Due to their production as IgG1 subclass, all six rAbs have the same Fc part; therefore, the differences in reactivity depend on the antigen recognition site only.

ELISA is a standard method to investigate antibody binding to PF4 and/or PF4/polyanion complexes in HIT as well as in VITT [22]. In the anti-PF4 ELISA and the anti-PF4/heparin ELISA, we observed a concentration-dependent reactivity profile of the six rAbs, however with some differences. The order of reactivity in both ELISAs was CR22050 > CR23004 > CR23005 > CR22066 > CR22046 > CR23003. It has to be taken into consideration that due to assay preparation, in both ELISAs, uncomplexed PF4 molecules are present, while only in the anti-PF4/heparin ELISA additional PF4/heparin complexes are present. In general, the OD values in the anti-PF4/heparin ELISA were slightly lower than in the anti-PF4 ELISA. This is explained by the competitive binding of heparin and VITT antibodies to PF4 [7,24]. Furthermore, rAbs CR22046, CR22050, and CR22066 were also tested in anti-PF4 ELISA in a previous publication by Wang et al. showing a comparable concentration-dependent effect but with CR22066 displaying the highest reactivity of the three tested rAbs [16]. This might be caused by a slightly different presentation of PF4 in the two assays. PF4 is very conformation-sensitive, and minor differences in the assay components may cause a change in structure [25].

To better differentiate between the binding of the rAbs to PF4 or PF4/polyanion complexes, we used the rapid anti-PF4 and anti-PF4/polyanion chemiluminescence assays [10]. The rapid anti-PF4 chemiluminescent assay displayed clear concentration-dependent binding of the rAbs. The order of reactivity was CR23004 > CR23005 > CR22050 > CR 22066 > CR22046 > CR23003. Similarly to the results from the ELISAs, the rAbs CR22046 and CR23003 showed lower reactivity with PF4 compared to the other rAbs. The most interesting differentiating characteristic among the six rAbs, their capability to bind to PF4/polyanion complexes, was seen only for CR23004, CR23005, and CR22050, however, at high concentrations, starting from 120 µg/mL. The binding of these antibodies to epitopes presented on uncomplexed PF4 and PF4/polyanion complexes may in part explain their different reactivities in the microtiter plate ELISAs and that some VITT patient sera react in both the rapid PF4 chemiluminescence assay and the rapid PF4/polyanion chemiluminescence assay [24]. The antibodies that also reacted with PF4/heparin complexes are also the ones showing the strongest reactivity in the rapid anti-PF4 assay. Heparin and VITT antibodies compete for the same binding site on PF4 [7]. One potential explanation for their reactivity in the PF4/heparin rapid test is that these high avidity antibodies may outcompete heparin from its binding site on PF4 at very high concentrations. Other alternative explanations might be the lack of fully occupied heparin-binding sites, true binding of these antibodies to PF4/heparin complexes, or unspecific binding. Despite the high antibody concentration needed, this cross-reactivity might be biologically relevant. Anti-PF4 antibodies in VITT patients reach a concentration range of 0.1–1% of the total IgG fraction [26] (approx. 6–170 µg/mL).

It is widely accepted that not all anti-PF4 antibodies are relevant. Only the antibodies able to activate platelets are clinically relevant. We, therefore, tested the rAbs in functional assays. In a similar way as classic VITT sera, rAbs induced platelet aggregation in the functional assay PIPA when PF4 was added. Platelet activation occurred via FcγRIIa and was blocked for all six rAbs by IV.3, another rAb that blocks FcγRIIa. The order of reactivity was CR22050 > CR23003 > CR22046 > CR23005 > CR22066 > CR23004. These different reactivities are not caused by the abovementioned cross-reactivity of the rAbs with PF4/polyanion complexes, as CR22003 and CR22046 did not show cross-reactivity. Notably, we observed that the reactivity of the antibodies in this platelet activation assay differs compared to antigen assays, i.e., CR23003, which showed the lowest reactivity with PF4 in the antigen binding assays, has one of the highest abilities to induce platelet activation. This might be caused by the different capabilities of the rAbs to form immune complexes with PF4 on the platelet surface in comparison to binding to immobilized PF4 in ELISA or in chemiluminescence assays, as previously shown for HIT antibodies [27]. Also, the differences in concentration dependency to activate platelets of CR22046, CR22050, and CR22066 are of interest and extend previous data [16].

There are only a little data available on the different reactivities of anti-PF4 antibodies in human sera [26]. Our findings of the different reactivities of the rAbs make it very likely that human anti-PF4 antibodies also have different binding reactivities depending on their complementarity-determining regions and variable region amino acid replacement mutations. This indicates that the different capabilities of anti-PF4 antibodies to cluster PF4 on the platelet surface and to activate platelets at different concentrations may explain why the clinical presentation of patients with the same reactivity of their antibodies in the antigen test differs in the severity of clinical complications. As a limitation of this study, not enough material remained of the original patient sera from which the rAbs were derived to be used as a control in the experiments. Moreover, the different IgG subclasses of the anti-PF4 antibodies in the patient sera may impact the correlation between the ELISA and functional assay results. The impact of IgG subclasses was not tested, as the rAbs are all IgG1 subclass.

VITT anti-PF4 antibodies were also found to promote procoagulant platelet activation [24,28,29]. Procoagulant platelets are a subset of highly activated platelets, which express P-selectin and phosphatidyl serine on their surface, facilitating the binding of coagulation factors [30,31]. All six rAbs increase procoagulant platelets compared to the control human IgG (CR22066 > CR23004 > CR23005 > CR22046 > CR23003 > CR22050). Although CR22050 and CR23003 showed the highest reactivity in PIPA, they showed the lowest potential in procoagulant platelet formation. In contrast, CR22066 and CR23004 showed higher induction of a procoagulant state but less strongly activated platelets in PIPA. Based on the results of this additional functional test, again, no clear evidence for a higher platelet activation potential of rAbs CR23004, CR23005, and CR22050, which also bind to PF4/polyanion complexes, could be observed. First clinical data from other diseases, e.g., ischemic stroke, hint towards an association of procoagulant platelet formation and adverse outcomes in patients [32]. Whether this is also relevant for the clinical severity of VITT requires in vivo animal experiments.

Together, the presented experiments reiterate that different factors might be in play in the biological effects caused by anti-PF4 antibodies, including antibody concentration, and avidity, potential cross-reactivity with PF4/polyanion complexes, reactivity in functional platelet activation assays, and the ability to generate a procoagulant platelet sub-population. Some of these characteristics and their impact can now be assessed in vivo in well-designed animal experiments using the presented rAbs. Particularly, these rAbs will be useful reagents for in vivo experiments to elucidate the mechanisms of thrombi formation in blood vessels at unusual sites. The use of these highly standardized antibodies will facilitate better characterizing the dose-dependent involvement of different cell populations and signaling cascades than the use of patient sera/plasma from plasmapheresis. Furthermore, the therapeutic potential of different drugs to treat VITT can be assessed in a more structured approach.

## 5. Conclusions

In conclusion, our results emphasize that patient-derived rAbs can be used in binding and platelet activation assays as a promising tool to study VITT and to unveil further details of this adverse vaccination effect in animal experiments. The rAbs might also become an important tool to identify the VITT-triggering vaccine component. This, in turn, would be a crucial step to understand the safety of adenoviral vaccines. rAbs can further be useful tools to standardize diagnostic tests to detect disease-causing antibodies in patient sera. Earlier reports exemplify the usage of 5B9, a monoclonal IgG antibody, in standardizing diagnostic tests to identify HIT [33,34]. Beyond VITT, a so-called VITT-like disorder is increasingly recognized. Several patients have been identified with this VITT-like disorder without previous vaccination, most likely triggered by a viral infection [10,35,36]. A perspective of the present study is to also generate rAbs from patients with this new VITT-like disorder to further assess similarities and differences between VITT and VITT-like disorders.

## Figures and Tables

**Figure 1 vaccines-13-00003-f001:**
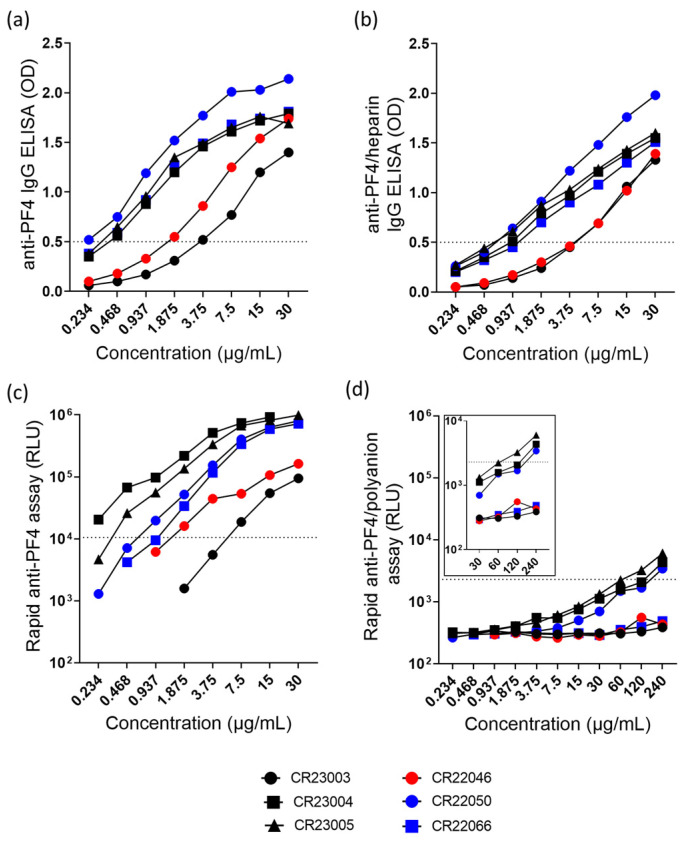
(**a**) Anti-PF4-IgG ELISA and (**b**) anti-PF4/heparin-IgG ELISA of the rAbs tested at concentrations ranging between 0.234 and 30 µg/mL. (**c**) Rapid anti-PF4 assay and (**d**) rapid anti-PF4/polyanion assay with the rAbs tested at concentrations ranging between 0.234 and 240 µg/mL. The insert represents the rAbs tested at higher concentrations (30–240 µg/mL). The range of the *y*-axis in the inset was limited to 10^4^ RLU for better visualization of the data points. The cut-off values of optical density (OD) and relative light units (RLU) are represented as dashed lines. Black: three different rAbs derived from patient P1; red: one rAb derived from patient P2; blue: two different rAbs derived from patient P3.

**Figure 2 vaccines-13-00003-f002:**
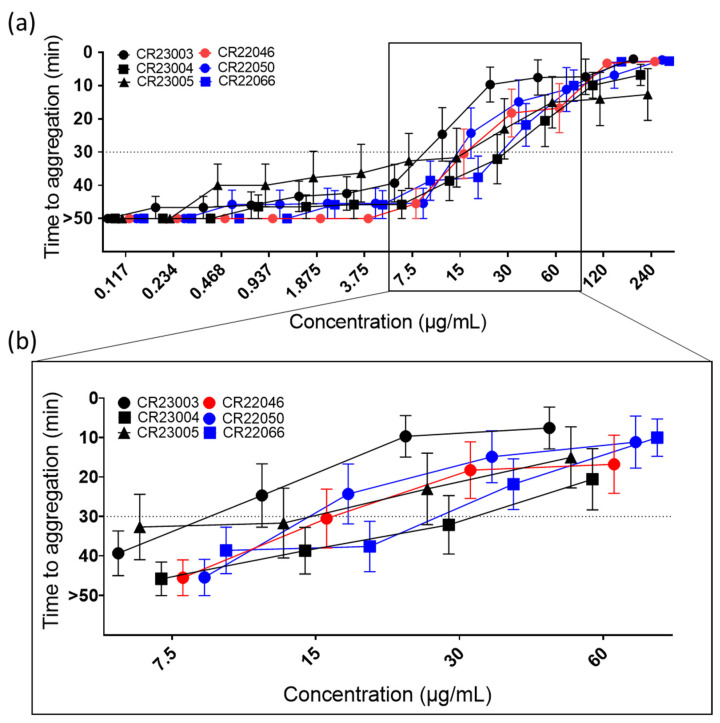
(**a**) Functional platelet activation assays (PIPA test) using recombinant antibodies (rAbs) at concentrations between 0.117 and 240 µg/mL. Cut-off at 30 min of the test is represented by the dashed line; platelets that were not activated within the 45 min observation time were assigned to 50 min (=no aggregation). The tests were carried out using 9 independent platelet donors. The results are given as mean ± SEM. For concentrations at which not all platelets are aggregated, means are calculated using the assigned 50 min for no aggregation. (**b**) For better visualization of critical concentrations, they are presented in a higher magnification. At 7.5 µg/mL, platelet aggregation was still below the cut-off for all the tested rAbs; at the next higher concentration CR23003 and CR22050 showed a positive PIPA result at 15 µg/mL, followed by CR23005, CR22046, and CR22066 at 30 µg/mL; and finally at 60 µg/mL all the tested rAbs showed a positive PIPA result. Black: three different rAbs derived from patient P1; red: one rAb derived from patient P2; blue: two different rAbs derived from patient P3.

**Figure 3 vaccines-13-00003-f003:**
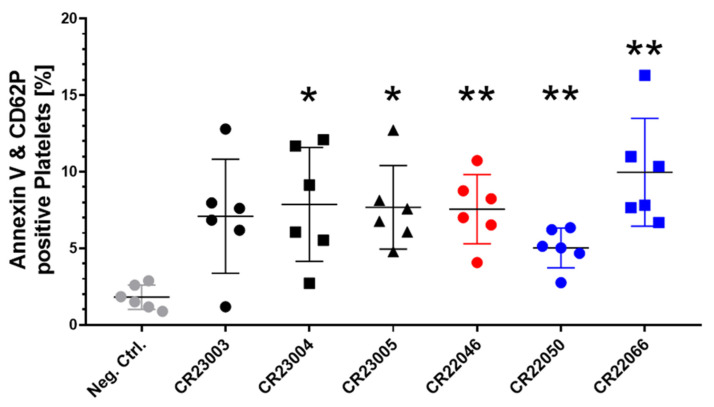
Quantitative analysis of recombinant antibody (rAb)-induced procoagulant platelets in the presence of externally added PF4, double positive for surface markers Annexin V and CD62P. All the rAbs were used at a concentration of 30 µg/ml and tested with washed platelets of six healthy donors. As a negative control (gray), we added human IgG instead of rAbs to the platelets. The results are given as mean ± SD. Data distribution was assessed using the Shapiro–Wilk test. Statistical analysis was performed by Brown–Forsythe and Welch ANOVA followed by Dunnett’s T3 test for multiple comparisons. * *p* < 0.05; ** *p* < 0.01 against the negative control. Black: three different rAbs derived from patient P1; red: one rAb derived from patient P2; blue: two different rAbs derived from patient P3.

**Table 1 vaccines-13-00003-t001:** Demographic and clinical features of patients with VITT.

Patient	Age	Sex	Diagnosis	Present IgG Subclass in Patient	Phenotype	Platelet Nadir	OD Anti-PF4-IgG ELISA	Derived Recombinant Anti-PF4 Antibodies (All Expressed as IgG 1)
P1 ^1^	44	F	Classic VITT after Ad26.COV2.S (Jcovden, Johnson & Johnson-Janssen)	IgG1	DVT, PE, PVT	14	3.57	CR23003 CR23004 CR23005
P2 ^2^	59	F	Classic VITT after ChAdOx1 nCoV-19 (Vaxzevria, AstraZeneca)	IgG1	CVST, PE, SVT	13	3.39	CR22046
P3 ^3^	49	F	Classic VITT after ChAdOx1 nCoV-19 (Vaxzevria, AstraZeneca)	IgG2	CVST, DVT	40	1.08	CR22050 CR22066

CVST = cerebral venous sinus thrombosis; DVT = deep vein thrombosis of legs; F = female; PE = pulmonary embolism; PVT = portal vein thrombosis; SVT = splanchnic vein thrombosis. ^1^ the data of patient P1 was previously published as part of the German VITT cohort [20]. The data of patients P2 and P3 was previously published as VITT2 ^2^ and VITT5 ^3^ in [19].

## Data Availability

The data underlying this publication will be made available upon written request to the corresponding author. The details of individual patients beyond the information given in this publication will not be made available according to data protection regulations.

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
