# Peer review of "Recombinant Anti-PF4 Antibodies Derived from Patients with Vaccine-Induced Immune Thrombocytopenia and Thrombosis (VITT) Facilitate Research and Laboratory Diagnosis of VITT"

_vaccines, 2024, doi:10.3390/vaccines13010003_

Round 1

Reviewer 1 Report

Comments and Suggestions for Authors

In this study, Dr. Wang and Dr. Schönborn joined to  produce recombinant PF4 antibodies from the serum of three VITT patients by identifying the sequence of the epitope-binding region through mass spectrometry fingerprinting. They then tested six selected clones for PF4 binding and platelet aggregation,

Minor comments:

Clarification of the overarching study aims: The authors state that the project aims to establish a foundation for future experiments to elucidate the mechanisms of VITT. It would be helpful to clearly define the specific open questions or hypotheses that also will be adressable by using the CR-antibodies.

Details of the recombinant antibody generation process: while the procedure for translating purified polyclonal PF4 antibodies into recombinant monoclonal antibodies is referenced in detail in reference 16, a more comprehensive description is needed for clarity. This is important as not all readers may have access to this article. Specifically, it is unclear: How many sequences were analyzed during the process. How many clones were initially identified. The criteria for selecting the six clones that were further studied for PF4 binding and platelet aggregation. Whether the six selected clones are part of the study in reference 16, and if so, what additional value or advantage is provided by including these extra six clones.

Lack of control comparison: The study would benefit from a comparison with relevant controls, such as purified polyclonal IgG from HIT patients, to strengthen the interpretation of the results.

Timing of serum sample collection: the timepoint at which the serum samples were collected for the purification of anti-PF4 antibodies matters and should be specified. Given that the half-life of pathogenic antibodies may be short following resolution of the clinical situation, it is crucial to mention whether the isolated antibodies represent those present during the acute phase of VITT or if they might differ from those in the acute setting.

Procoagulant platelet measurement by FC: In the results section (or the figure caption), it should be explicitly stated that PF4 was added during the measurement of procoagulant platelets. As written, it may be interpreted that the CR antibodies alone induced platelet activation. Additionally,  the use of washed platelets and application of shear stress are mentioned, further details are needed: How was shear stress applied? What was the magnitude and duration of the shear stress, and the rationale for its application?

 Chemiluminescence test description: The description of the chemiluminescence test currently explains the first step, where the recombinant antibodies (rABs) bind to PF4-coated magnetic particles. However, the second step, as well as the readout process, are not sufficiently described. A more detailed explanation of the full procedure and how the results are quantified would be beneficial.

Author Response

Point-by-Point Reply to Reviewer 1

In this study, Dr. Wang and Dr. Schönborn joined to produce recombinant PF4 antibodies from the serum of three VITT patients by identifying the sequence of the epitope-binding region through mass spectrometry fingerprinting. They then tested six selected clones for PF4 binding and platelet aggregation.

Response: We thank the reviewer for the constructive comments and would like to answer to them as follows:

1. Clarification of the overarching study aims: The authors state that the project aims to establish a foundation for future experiments to elucidate the mechanisms of VITT. It would be helpful to clearly define the specific open questions or hypotheses that also will be adressable by using the CR-antibodies.

Response:
Hypotheses that will be addressable by using the described rAbs are added in more detail to the discussion section. (page 12):

“Together, the presented experiments reiterate that different factors might be in play in the biological effects caused by anti-PF4 antibodies. Antibody concentration, and avidity, potential cross-reactivity with PF4/polyanion complexes, reactivity in functional platelet activation assays, and the ability to generate a procoagulant platelet sub-population. Some of these characteristics and their impact can now be assessed in vivo in well-designed animal experiments using the presented rAbs. Particularly these rAbs will be useful reagents for in vivo experiments to elucidate the mechanisms on thrombi formation in blood vessels at unusual sites. The use of these highly standardized antibodies will facilitate to better characterize dose dependent involvement of different cell populations and signaling cascades than the use of patient sera/plasma from plasmapheresis. Furthermore, the therapeutic potential of different drugs to treat VITT can be assessed in a more structured approach.”

2. Details of the recombinant antibody generation process: while the procedure for translating purified polyclonal PF4 antibodies into recombinant monoclonal antibodies is referenced in detail in reference 16, a more comprehensive description is needed for clarity. This is important as not all readers may have access to this article. Specifically, it is unclear: How many sequences were analyzed during the process. How many clones were initially identified. The criteria for selecting the six clones that were further studied for PF4 binding and platelet aggregation. Whether the six selected clones are part of the study in reference 16, and if so, what additional value or advantage is provided by including these extra six clones.

Response:
More details on the recombinant antibody generation process have now been added to the method (lines 116-122, page 3):

“DNA fragments encoding full length IgG antibody heavy chain (HC) and light chain (LC) were synthesized and cloned into pcDNA3.4 vector separately, in frame with an artificial signal peptide in front. Resulted constructs were co-transfected into CHO cell at a HC:LC ratio of 1:1. The culture medium supernatant was harvested by ultrafiltration, then subject to Protein A affinity purification and SEC-HPLC chromatography polishing. Antibody purity was characterized by SDS-PAGE. The purified antibodies were dialyzed into 20 mM NaAC, 75 mM NaCl, 5 % sucrose, pH 5.5 and stored at -80 °C until required.”

In our previous work (Ref19 in the manuscript), mass spectrometric sequencing of anti-PF4 antibodies showed a single IgG heavy chain species paired with a single lambda light chain species in unrelated patients. Amino acid replacement mutations were found in individual patient’s heavy chain and light chain variable regions, indicating there were intraclonal variants (Ref19). In this current manuscript, six recombinant anti-PF4 antibodies were derived from three different patients with classic VITT after vaccination with ChAdOx1 nCoV-19 (P2 and P3) or Ad26.COV2.S vaccine (P1). CR22046 (derived from P2) and CR22050 and CR22066 (derived from P3) were characterized for their PF4 binding, platelet aggregation, epitope mapping and paratope modeling in reference 16. However, in the current study, these three recombinant antibodies were further characterized for their PF4 binding and platelet aggregation capacities by using different sets of immunoassays as described in the methods. More importantly, the recombinant antibodies of CR23003, CR23004 and CR23005 were derived from VITT patient P1 after vaccination with Ad26.COV2.S vaccine. As anti-PF4 response after Ad26.COV2.S has not been previously studied, these antibodies are very valuable research reagents to study VITT pathogenesis in comparison with ChAdOx1 nCoV-19-triggered VITT. Therefore, it is important to characterize the recombinant antibodies from patients after both adenoviral-vectored COVID-19 vaccines to evaluate their utility to study VITT pathophysiology.

3. Lack of control comparison: The study would benefit from a comparison with relevant controls, such as purified polyclonal IgG from HIT patients, to strengthen the interpretation of the results.

Response:
We apologize for not being clear enough in the manuscript. HIT anti-PF4 antibodies are different to VITT anti-PF4 antibodies and react differently in the assays. They would not be a valid control. The antibodies in VITT patient sera are also mostly monoclonal or at best oligoclonal. Therefore, the sera react very similar to the monoclonal antibodies. Unfortunately, the corresponding sera of the patients from whom we have purified the antibodies to generate the recombinant antibodies are nearly used up for these experiments. We included this as a potential study limitation in the discussion section. (page 11)

“This indicates that the different capabilities of anti-PF4 antibodies to cluster PF4 on the platelet surface and to activate platelets at different concentrations may explain why the clinical presentation of patients with same reactivity of their antibodies in the antigen test differ in the severity of clinical complications. As a limitation of this study, of the original patient sera from which the rAbs were derived not enough material remained to be used as controls in the experiments. Moreover, the different IgG subclasses of anti-PF4 antibodies in patient sera may impact the correlation between ELISA and functional assay results. The impact of IgG subclasses was not tested, as the rAbs are all IgG1 subclass.”

4. Timing of serum sample collection: the timepoint at which the serum samples were collected for the purification of anti-PF4 antibodies matters and should be specified. Given that the half-life of pathogenic antibodies may be short following resolution of the clinical situation, it is crucial to mention whether the isolated antibodies represent those present during the acute phase of VITT or if they might differ from those in the acute setting.

Response: We totally agree with the reviewer that the timing of serum sample collection is critical and thank the reviewer to make us aware to include this information into the manuscript. The serum samples from all patients were collected within the acute phase after onset of symptoms (P1 = 7 days after vaccination, P2 = 15 days after vaccination, and P3 = 12 days after vaccination, respectively).
We have added the time point of sample collection to the methods section (page 3) as this is a highly relevant aspect for the relevance of the recombinant antibodies to recapitulate acute VITT.

“Serum samples with suitable volumes of three VITT patients from the acute phase (obtained 7-15 days after vaccination) were used to affinity-purify serum anti-PF4 antibodies as previously described [19].”

5. Procoagulant platelet measurement by FC: In the results section (or the figure caption), it should be explicitly stated that PF4 was added during the measurement of procoagulant platelets. As written, it may be interpreted that the CR antibodies alone induced platelet activation. Additionally, the use of washed platelets and application of shear stress are mentioned, further details are needed: How was shear stress applied? What was the magnitude and duration of the shear stress, and the rationale for its application?

Response:
This additional information is now added to the legend of Figure 3 as well as to the Materials and Methods section (p. 5).

To study the ability of the rAbs to cause a procoagulant state in platelets on the addition of PF4, flow cytometry was performed using washed platelets of six different healthy donors. Washed platelets were preincubated with ReoPro (1:500, Eli Lilly, Giessen, Ger-many, blocking the platelet fibrinogen receptor) for 5 min to avoid platelet aggregation that would compromise subsequent measurements. To 35 µL pretreated platelets 10 µL rAbs (30 µg/mL) or human IgG isotype control (30µg/mL, Cat. No.: 12000C, Invitrogen, Rockford, IL, USA) and 6µL PF4 (100 µg/mL, PF4-h, Chromatec, Greifswald, Germany) were added and incubated under shear stress. Shear stress was applied for 5 min using two magnetic stirring beads added to each well of a 96-well plate placed on a magnetic laboratory stirrer (IKA C-MAG MS 7 at speed 2 = ~500 rpm). Then an antibody master mix was added to the platelets, consisting of: 68.4 µL Annexin V binding buffer (Cat. No.: 422201, BioLegend, Amsterdam, The Netherlands), 1.6 µL Hirudin 5400 U, 5 µL Annexin V (Cat. No.: A35110, APC-labelled, Bio-Legend, Amsterdam, The Netherlands), 5 µL CD62P-FITC (Cat. No: A07790, Beckmann Coulter, Villepinte, France) and 10 µL CD42b-PE (Cat. No: IM1417U, Beckmann Coulter, Villepinte, France). Platelets were incubated for 20 min at room temperature in the dark and measured using CytoFlexS (Beckmann Coulter, Villepinte, France). Events were pre-gated for the platelet-marker CD42b and platelets positive for both, CD62P and for the phosphatidylserine marker Annexin V, were quantified.

6. Chemiluminescence test description: The description of the chemiluminescence test currently explains the first step, where the recombinant antibodies (rABs) bind to PF4-coated magnetic particles. However, the second step, as well as the readout process, are not sufficiently described. A more detailed explanation of the full procedure and how the results are quantified would be beneficial.

Response: We added additional information on the second step and the read-out process to the Methods Section (page 4).

“Both assays are two-step chemiluminescence immunoassays consisting of magnetic particles coated either with PF4 complexed to PVS or with PF4 alone. rAbs bind to the PF4 or PF4/PVS coated magnetic particles, and are detected using an anti-human IgG labelled with isoluminol. Both assays are used in the ACL AcuStar chemiluminescence analyzer (from Werfen, Barcelona Spain).”

Reviewer 2 Report

Comments and Suggestions for Authors

This interesting a paper about Covid-19, it could be published but several minor changes need to be done.

1)    The authors indicate that adenoviral vector-based vaccines against COVID-19 rarely cause vaccine-induced immune thrombocytopenia and thrombosis (VITT). It would be interesting if they included some quantitative data. To situate the magnitude of the issue.

2)    In the material and methods, it should be mentioned how many patients were sampled.

3)      In the material and methods the authors coment that they used ROUT method for outlier identification and removal in your analysis. Please Explain the acronym ROUT stands for Robust Regression and Outlier Removal

4)      In table 1, in the Patient column, there are superscripts 1, 2, and 3 but they are not explained at the bottom of the table. It seems that only superscript 1 is

Author Response

Point-by-Point Reply to Reviewer 2

This interesting a paper about Covid-19, it could be published but several minor changes need to be done.

Response: We thank the reviewer for the important comments and would like to answer as follows:

1. The authors indicate that adenoviral vector-based vaccines against COVID-19 rarely cause vaccine-induced immune thrombocytopenia and thrombosis (VITT). It would be interesting if they included some quantitative data. To situate the magnitude of the issue.

Response:
We agree with the reviewer and added information on the incidence to the Introduction. (page 2)

“The incidence differs depending on the reporting country and the vaccine used. Incidence ranges from 1 case per 26,500 to 127,300 for the first doses of ChAdOx1 nCoV-19 and has been estimated as 1 case per 263,000 Ad26.COV2.S doses administered in the US [3].”

2. In the material and methods, it should be mentioned how many patients were sampled.

Response:
We added this information to the Methods Section.

“Serum samples with suitable volumes of three VITT patients from the acute phase (obtained 7-15 days after vaccination) were used to affinity-purify serum anti-PF4 antibodies as previously described [19]”

3. In the material and methods the authors coment that they used ROUT method for outlier identification and removal in your analysis. Please Explain the acronym ROUT stands for Robust Regression and Outlier Removal

Response:
The acronym is now explained accordingly.

4. In table 1, in the Patient column, there are superscripts 1, 2, and 3 but they are not explained at the bottom of the table. It seems that only superscript 1 is

Response:
The missing superscripts 2 and 3 were added to the explanation at the bottom of the table 1.

Reviewer 3 Report

Comments and Suggestions for Authors

Müller et al., have submitted the short study entitled “Recombinant anti-PF4 antibodies derived from patients with vaccine-induced immune thrombocytopenia and thrombosis (VITT) facilitate research and laboratory diagnosis of VITT”.  

The authors would like to delineate the function of the anti-PF4 antibodies on human platelets. Due to the lack of monoclonal antibodies against PF4, the authors used the mass spec sequencing facility to generate recombinant monoclonal antibodies for this study. The author generated 6 different mAbs against PF4 and evaluated their binding affinity and platelet agglutinating functionality.

This reviewer has the following comments:

Why were only six antibodies considered, despite the knowledge that several CDRs could potentially be effective against PF4?

In my experience, handling platelets can be challenging due to their rapid activation following a brief stimulus, such as vigorous shaking. As activation leads to functional changes, which may give false positive/negative results.

There are very special reagents and antibodies; please provide the catalogue numbers for reproducibility.

In addition, the authors have not presented the sequence information in the supplementary. The main manuscript should represent a comparison of 6 antibody protein sequences.

Figure 2: I do not see any significant change in the statistics. As you can see, despite significant changes in binding, there are no significant differences in antibody activation.

Figure 3: Statistics should be explained. Compared with what?

The major drawback of this study is the lack of in vivo validation in a mouse model.

Author Response

Point-by-Point Reply to Reviewer 3

Müller et al., have submitted the short study entitled “Recombinant anti-PF4 antibodies derived from patients with vaccine-induced immune thrombocytopenia and thrombosis (VITT) facilitate research and laboratory diagnosis of VITT”.  

The authors would like to delineate the function of the anti-PF4 antibodies on human platelets. Due to the lack of monoclonal antibodies against PF4, the authors used the mass spec sequencing facility to generate recombinant monoclonal antibodies for this study. The author generated 6 different mAbs against PF4 and evaluated their binding affinity and platelet agglutinating functionality.

Response: We thank the reviewer for the constructive comments and would like to answer to them as follows:

1. Why were only six antibodies considered, despite the knowledge that several CDRs could potentially be effective against PF4?

Response: As shown in our previous work (Ref19), VITT anti-PF4 antibodies are highly structurally similar across unrelated patients. These unique serum clonotypes are specified by identical IGLV3-21*02 allelic light-chains paired with shared heavy and light CDR3 amino acids motifs. Therefore, we consider that 6 different antibodies derived from 3 different patients after two different adenoviral-vector vaccines should be sufficient given the similarities between the hypervariable regions of these antibodies.

2. In my experience, handling platelets can be challenging due to their rapid activation following a brief stimulus, such as vigorous shaking. As activation leads to functional changes, which may give false positive/negative results.

Response:
We fully agree with the notion that platelets are difficult to handle. Our laboratory has more than 30 years expertise and daily routine in working with these cells, both in research and diagnostics. Generation of platelets was performed using in-house SOPs with suitable quality controls. We can exclude artefacts relevant for the experiments described in the manuscript.

3. There are very special reagents and antibodies; please provide the catalogue numbers for reproducibility.

Response:
The missing information was added to the text in the Materials and Methods section, whenever possible. However, for the rAbs as well as the rapid anti-PF4 chemiluminescence assay this cannot be provided, as they are not commercially available at the moment.

4. In addition, the authors have not presented the sequence information in the supplementary. The main manuscript should represent a comparison of 6 antibody protein sequences.

Response:
Unfortunately the presentation of the protein-sequences of the rAbs to the broad reader audience is currently not possible due to confidentiality reasons with intellectual property regulations and other upcoming manuscripts in preparation. Researchers interested in this information may contact us and we are prepared sharing this information on the basis of bilateral information transfer agreements.

5. Figure 2: I do not see any significant change in the statistics. As you can see, despite significant changes in binding, there are no significant differences in antibody activation.

Response:
We fully agree with the notion of the reviewer. With this figure we wanted to emphasize the general ability of the rAbs to activate platelets in the functional PIPA test, similar to previous reports of patient sera. There was only a concentration-dependency shown by statistical testing, but no statistically significant difference between the six tested rAbs. The Results section (3.2.2, p.8) was revised to clarify this to the reader.

“All rAbs activated platelets in a concentration-dependent manner (Figure 2a, ordinary 2-way ANOVA, F (11, 516)=79.35, p<0.0001) in the PIPA test with added PF4. Activation-capability differed among antibodies, with activation thresholds between 15 µg/mL and 60 µg/mL (Figure 2b). However, the differences in activation capability were not statistically significant. Using human IgG isotype as a negative control, no reactivity in the test was observed.”

6. Figure 3: Statistics should be explained. Compared with what?

Response:
Due to normally distributed data with unequal standard deviations, testing was performed as a Brown-Forsythe and Welch ANOVA with multiple comparisons against the negative control, followed by a Dunnet´s T3 correction as recommended for smaller sample sizes per group. The information of the testing against the negative control group was added to the legend of Figure 3.

7. The major drawback of this study is the lack of in vivo validation in a mouse model.

Response:
We agree with the reviewer that the presented rAbs should be tested in an in vivo mouse model. This point is now discussed in further detail in the Discussion Section (p. 12). However, these experiments are ongoing work and would be beyond the scope of the present article. The dilemma of animal experiment approval by our authorities is that only validated reagents must be used to reduce unnecessary mouse experiments. The present manuscript provides this validation.

“[…] Some of these characteristics and their impact can now be assessed in vivo in well-designed animal experiments using the presented rAbs. Particularly these rAbs will be useful reagents for in vivo experiments to elucidate the mechanisms on thrombi formation in blood vessels at unusual sites. The use of these highly standardized antibodies will facilitate to better characterize dose dependent involvement of different cell populations and signaling cascades than the use of patient sera/plasma from plasmapheresis. Furthermore, the therapeutic potential of different drugs to treat VITT can be assessed in a more structured approach.”

Round 2

Reviewer 3 Report

Comments and Suggestions for Authors

Thank you for the revised version. I recommend the revised version as a brief communication.